# Understanding hydrogen-bonding structures of molecular crystals via electron and NMR nanocrystallography

Candelaria Guzmán-Afonso [1,9], You-lee Hong [1,2,9], Henri Colaux[1], Hirofumi Iijima[3], Akihiro Saitow[3], Takuma Fukumura[3], Yoshitaka Aoyama[3], Souhei Motoki[3], Tetsuo Oikawa[4], Toshio Yamazaki[5], Koji Yonekura[6,7] & Yusuke Nishiyama [1,5,8]

Understanding hydrogen-bonding networks in nanocrystals and microcrystals that are too small for X-ray diffractometry is a challenge. Although electron diffraction (ED) or electron 3D crystallography are applicable to determining the structures of such nanocrystals owing to their strong scattering power, these techniques still lead to ambiguities in the hydrogen atom positions and misassignments of atoms with similar atomic numbers such as carbon, nitrogen, and oxygen. Here, we propose a technique combining ED, solid-state NMR (SSNMR), and first-principles quantum calculations to overcome these limitations. The rotational ED method is first used to determine the positions of the non-hydrogen atoms, and SSNMR is then applied to ascertain the hydrogen atom positions and assign the carbon, nitrogen, and oxygen atoms via the NMR signals for $^1$H, $^{13}$C, $^{14}$N, and $^{15}$N with the aid of quantum computations. This approach elucidates the hydrogen-bonding networks in L-histidine and cimetidine form B whose structure was previously unknown.

[1] Nano-Crystallography Unit, RIKEN-JEOL Collaboration Center, Tsurumi, Yokohama, Kanagawa 230-0045, Japan. [2] Institute for Integrated Cell-Material Sciences (WPI-iCeMS), Institute for Advanced Study, and AIST-Kyoto University Chemical Energy Materials Open Innovation Laboratory (ChEM-OIL), Kyoto University, Yoshida, Sakyo-ku, Kyoto 606-8501, Japan. [3] JEOL Ltd., Musashino, Akishima, Tokyo 196-8558, Japan. [4] JEOL ASIA Pte. Ltd, Corporation Road, Singapore, Singapore. [5] NMR Science and Development Division, RIKEN SPring-8 Center, Tsurumi, Yokohama, Kanagawa 230-0045, Japan. [6] Biostructural Mechanism Laboratory, RIKEN SPring-8 Center, 1-1-1 Kouto, Sayo, Hyogo 679-5148, Japan. [7] Advanced Electron Microscope Development Unit, RIKEN-JEOL Collaboration Center, 1-1-1 Kouto, Sayo, Hyogo 679-5148, Japan. [8] JEOL RESONANCE Inc., Musashino, Akishima, Tokyo 196-8558, Japan. [9] These authors contributed equally: Candelaria Guzmán-Afonso, You-lee Hong. Correspondence and requests for materials should be addressed to Y.N. (email: yunishiy@jeol.co.jp)

Hydrogen bonding between hydrogen and electronegative atoms has key roles in the stabilization of inter- and intramolecular packing and the functions of molecules. To understand complex hydrogen-bonding networks in molecular crystals, it is of critical importance to elucidate their structures at atomic resolution, including the positions of the hydrogen atoms. Information regarding these structures is valuable for not only materials science but also pharmaceutical research and biology. Thus, typical target molecules include pharmaceutical compounds, metal–organic frameworks, peptides, etc. If a large single crystal (> 10–100 μm) of sufficiently high quality is available, either single-crystal (SC) X-ray diffraction (XRD) or neutron diffraction (ND) can be used to determine the structure, including that of the hydrogen-bonding networks. In contrast, single crystals with nanometer to micrometer dimensions seldom yield diffraction spots even using high-intensity X-rays from a modern synchrotron source. Although powder XRD (PXRD) can be used if a large amount (ca. 1 mg) of such small crystals is available, this technique does not readily permit localization of the hydrogen atoms and also requires highly isomorphic microcrystalline samples with small lattices. Larger lattices may result in overlap of the powder diffraction patterns and prevent the extraction of accurate intensities. Furthermore, PXRD often suffers from ambiguity in the identification of atoms with similar atomic numbers, such as carbon, nitrogen, and oxygen atoms, rendering the structure determination of organic molecules difficult. Although the presence of hydrogen bonds in nanocrystals and microcrystals can be determined using infrared (IR) spectroscopy, the peak positions are strongly affected by external parameters, such as temperature, pressure, and concentration, and overlapping peaks may further complicate the analysis. Solid-state nuclear magnetic resonance (SSNMR) provides valuable information regarding hydrogen bonding through both isotropic and anisotropic chemical shift values[1–3], proximities between hydrogen atoms, and $^1H–X$ (X = $^{13}C$, $^{15}N$) internuclear distance measurements[4,5]. Nevertheless, whole-crystal structures can seldom be obtained solely using IR and SSNMR. Thus, the combined use of XRD, SSNMR, and quantum computation has proved to be a powerful approach for understanding crystal structures and hydrogen-bonding networks as it can overcome the limitations of each individual technique[6–11]. In this approach, XRD is used to determine all of the atomic positions in a crystal except for those of the hydrogen atoms, and then quantum computation is used to place the missing hydrogen atoms in reasonable positions, which can be subsequently verified using SSNMR. Moreover, as SSNMR permits direct measurements of carbon and nitrogen atoms, it removes uncertainties in the atomic assignments, an inherent difficulty of PXRD analysis. In particular, recent progress in fast magic-angle sample spinning (MAS) SSNMR techniques has enabled the direct measurement of hydrogen atoms, and quantum computation can be used to link crystal structures to NMR chemical shifts. As the information provided by SSNMR is complementary to that obtained using XRD, this approach, which is referred to as NMR crystallography, has become popular for the structural determination of SC or pure powder samples. However, there remain several obstacles that limit the widespread application of this approach. When samples form as nanocrystals or contain multiple components such as in the case of pharmaceutical tablets, it is difficult to solve the molecular structure using XRD, which hampers the NMR crystallography approach. In such cases, electron diffraction (ED) or electron 3D crystallography can be applied to determine the nanocrystal structure. ED permits the three-dimensional (3D) structures to be solved through 3D reciprocal space using diffraction tomography and continuous rotation methods[12–22]. Owing to the much stronger interaction of electrons with the

sample compared with X-rays, the nanocrystals afford clear diffraction spots suitable for structure determination. It was recently demonstrated that the structures of nanocrystals including the hydrogen positions could be determined solely using ED by considering the influence of dynamic scattering during data analysis[12]. In these ideal cases where all of the hydrogen atoms are visible in the ED potential maps, the hydrogen atoms can be located with sufficient accuracy to understand the hydrogen-bonding network. However, even with this advanced approach, dynamical refinement does not always permit localization of all of the hydrogen atoms and moreover is not applicable to samples susceptible to radiation damage and/or possessing complex structures[23,24]. Furthermore, the accuracy of the hydrogen positions is not sufficiently high to determine the hydrogen-bonding strength, which directly influences the stabilization mechanism and is a major issue in engineering, medicine, and materials science. Moreover, the unambiguous identification of atoms with similar atomic numbers remains problematic in ED methods without dynamical refinements.

Herein, we demonstrate the use of a combined approach involving ED, SSNMR, and quantum computation to determine the structures of nano- to microcrystalline organic compounds together with their hydrogen-bonding networks. The nanocrystal structure of the organic samples is first determined via ED using the rotation method. The ambiguities originating from both misassignment of carbon, nitrogen, and oxygen atoms and missing or ambiguous hydrogen atoms are then solved via SSNMR combined with quantum computation based on NMR crystallography. We first report on the application of this approach to the model system of pure orthorhombic L-histidine, whose molecular structure is already known. The crystal and hydrogen-bonding structure are then solved for cimetidine form B (CB), whose structure has not previously been reported. This latter system is more challenging than L-histidine as CB recrystallizes in smaller crystals than required for SCXRD and is contaminated by different crystal forms, which precludes the use of SCXRD or PXRD for structure determination.

## Results

**Demonstration of the combination of ED, SSNMR, and GIPAW.** The initial molecular structure of an orthorhombic L-histidine microcrystal (Fig. 1a) was solved from ED patterns that were collected by continuously rotating the microcrystal around a single axis under electron irradiation (Fig. 1b). In principle, the entire torous in the 3D reciprocal space can be reconstructed from a single set of ED patterns. However, the limited range of sample rotation around the single axis covered only part of the reciprocal space, resulting in a missing wedge in the reconstructed reciprocal space. To cover this missing wedge, five sets of diffraction patterns from five crystals were combined using the program BLEND in the Collaborative Computational Project Number 4 software[25]. Each set of diffraction data was processed using the XRD crystallographic data processing software (XDS)[26]. Prior to merging the different sets of diffraction patterns, data with significantly different lattice parameters were excluded; this step is particularly important if the sample is contaminated (see the example of cimetidine below). The details of the data sets are summarized in Supplementary Table 1. The merged data for L-histidine provided the lattice parameters of $a = 5.27(5)$ Å, $b = 7.44(5)$ Å, and $c = 18.99(8)$ Å, the space group of $P2_12_12_1$, and orthorhombic crystal symmetry (see Supplementary Table 2). The molecular structure of the unit cell was solved via the standard direct method using the SIR2014 software[27]. However, the solved molecule contained misassigned carbon, nitrogen, and oxygen atoms and only three of the nine hydrogen atoms were identified

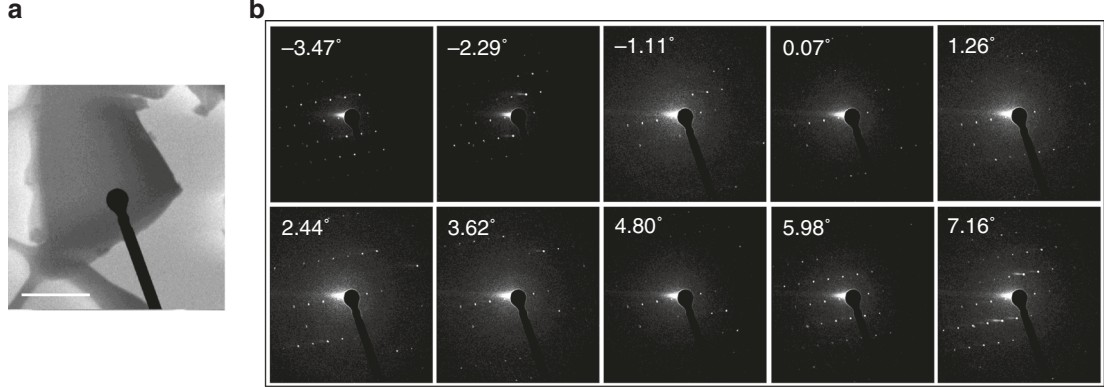

**Fig. 1** TEM image and a series of ED patterns of an L-histidine microcrystal. **a** TEM image of an L-histidine microcrystal on a ultra-thin carbon film obtained immediately after the ED measurement. **b** Selected ED patterns of the L-histidine microcrystal during continuous rotation. The number of each frame represents a starting rotation angle of each ED patterns. The diffraction patterns were obtained every 1.18°

(Fig. 2a). Furthermore, the positions of even the detected hydrogen atoms were revealed to be inaccurate in difference potential maps owing to the limited scattering power of hydrogen atoms (see Supplementary Figure 1). As the molecular structure of L-histidine is already known, the positions of the carbon, nitrogen, and oxygen atoms were easily refined and the hydrogen atoms were simply removed using the SHELXL software[28]. Nevertheless, ambiguities still remained because 180° flipping of the imidazole ring could not be distinguished owing to the lack of assignments for the imidazole carbon and nitrogen atoms, affording two candidate structures for L-histidine (LH), namely, LH1(2) (Fig. 2b, c) and LH3(4) (Fig. 2d, e), with R factors of 22.71% and 23.86%, respectively (see Supplementary Figure 1). Although the larger R factor and abnormal atomic displacement probability (see Supplementary Figure 1) for the LH3(4) models could exclude the possibility of these being the correct structure, more definitive confirmation was needed. The difference of 1.15% in the R factors of LH1(2) and LH3(4) indicates the limitation of ED in distinguishing atoms with similar atomic numbers. The two candidates were further complicated by the hydrogen atom positions; the protonation states of the two imidazole nitrogen atoms were unclear. Furthermore, it was not clear whether the molecule was zwitterionic. As these structures differ solely in terms of the hydrogen atom positions, it was crucial to correctly determine the locations of the hydrogen atoms. SSNMR partially addresses these questions via the $^1$H, $^{14}$N, and $^{15}$N signals, which can be easily observed, especially under very fast MAS conditions[29]. Both $^1$H–$^{14}$N and $^1$H–$^{15}$N 2D SSNMR spectra contained two covalently bonded N–H signals from the α-amino group and imidazole ring (see Supplementary Figure 2), which clearly indicated that only one of the two nitrogen atoms of the imidazole ring was protonated[30]. The peak at − 280 ppm in the $^{14}$N dimension empirically suggested the presence of $NH_3^+$, as small quadrupolar couplings with high symmetry in $^{14}NH_3^+$ give rise to a small second-order quadrupolar shift of $^{14}$N[30]. Thus, it was safe to conclude that L-histidine existed as a zwitterion. By combining these results, four plausible structures of LH1-4 can be considered (Fig. 2b–e). The hydrogen atoms were manually added using SHELXL with default bond lengths at the ED measurement temperature of 96.15 K: N2–H2A = N3–H3 = 0.88 Å, N1–H1A/B/C = 0.91 Å, C2–H2 = 1.0 Å, C3–H3A/B = 0.99 Å, C5–H5 = C6–H6 = 0.95 Å. Addition of the hydrogen atoms improved the R factors to between 20.09% and 21.28% for all of the structures. The R factors after each refinement step are plotted in Fig. 2f and summarized in Supplementary Table 3. Although the difference potential maps were not sufficient for determining all of the hydrogen atom positions (see Supplementary Figure 1),

this large decrease in the R factors demonstrates the importance of the presence of the hydrogen atoms during the structure refinement process. Nevertheless, the R factors did not reveal whether LH1 or LH2 was the correct structure even if LH3 and LH4 were excluded. The default bond lengths given by SHELXL tend to differ from the actual bond lengths as they do not take factors of length variation into account. $^1$H–X distances (X = $^{13}$C, $^{15}$N) can be more accurately measured using $^1$H–X 2D inversely proton-detected cross-polarization with variable contact time (invCP-VC) spectra, whose robustness and accuracy have been experimentally and theoretically demonstrated[4,31]. As the magnitude of the dipolar interaction is inversely proportional to the cube of the internuclear distance, the $^1$H–X dipolar interactions measured using invCP-VC SSNMR were directly converted to bond lengths using Equation (1) in the Methods section. In the 2D $^1$H–$^{13}$C and $^1$H–$^{15}$N invCP-VC spectra of L-histidine, the separation between two peaks in the $^1$H–X dimension represents the size of the dipolar interaction for each $^1$H–X pair (see Fig. 2g, h). The bond lengths were measured to be 1.12 Å (± 0.02 Å) for C2–H2, C5–H5, and C6–H6 and 1.07 Å (± 0.02 Å) for N3–H3 and N2–H2A. The N–H distances of the $NH_3^+$ moiety were not evaluated owing to the technical difficulty of determining small dipolar couplings of $NH_3^+$, which were dynamically scaled by a factor of ∼ − 0.33[32]. All of the structures were finally refined with the bond lengths determined using SSNMR. However, this procedure did not significantly improve the R factors, showing the insensitivity of ED measurements to the hydrogen positions. After adjusting the weights during the final step of structure refinement, the R factors decreased to 19.81%, 20.00%, 21.06%, and 21.76% for the LH1, LH2, LH3, and LH4 models, respectively. Although the LH1 conformation afforded the lowest R factor, that for LH2 was similar. These results demonstrate that the use of R factors alone is not sufficient for identifying the correct structure. It should be noted that we did not consider the influence of multiple scattering on the R factor; thus, the small variation of the R factor may originate from an incorrect structure and/or errors in evaluating the scattering intensities.

To identify the correct structure, gauge including projected augmented wave (GIPAW) calculations[33,34] and SSNMR were combined according to the NMR crystallography approach. GIPAW optimizes a crystal structure by relaxing the atomic positions to local minima of the energy surface in addition to providing NMR parameters such as isotropic chemical shifts and quadrupolar couplings. It should be noted that the GIPAW calculations were performed at 0 K (see Supplementary Table 4), whereas the SSNMR experiments were conducted at room temperature. Although the calculated energy minima sometimes

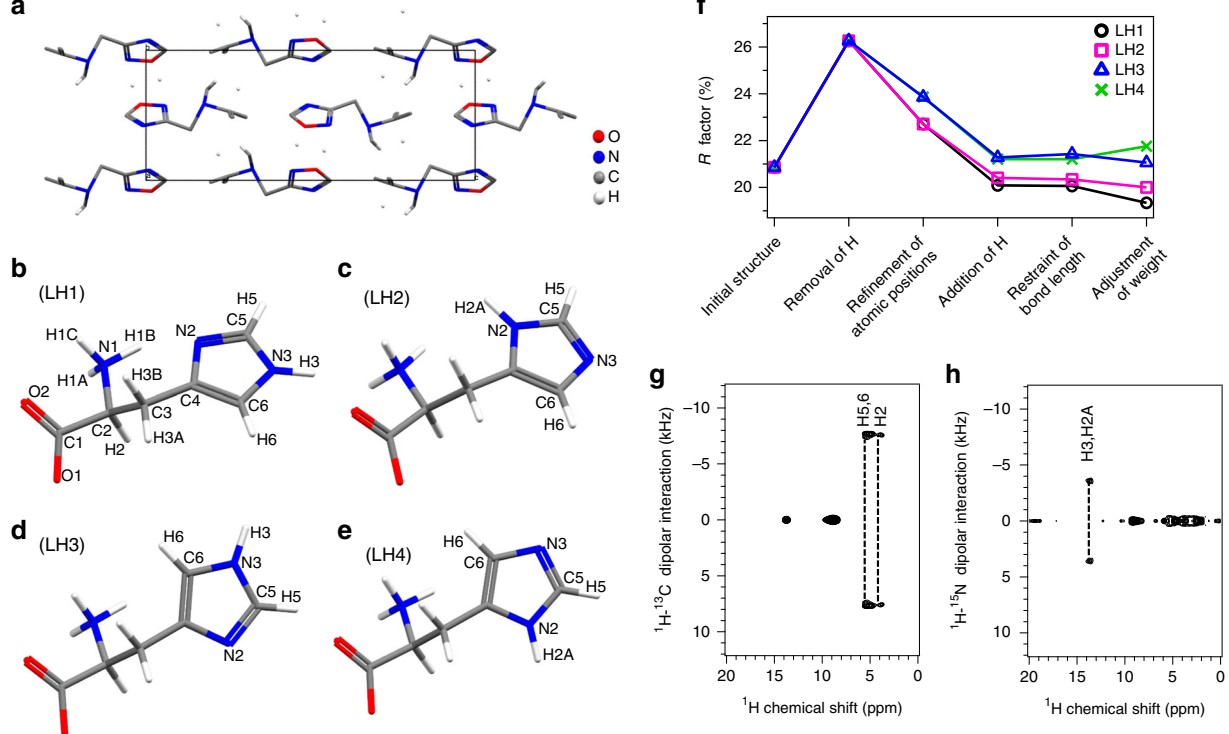

**Fig. 2** Determination of L-histidine crystal structure using ED and SSNMR. **a** The initial structure was determined via ED using the rotation method. The structure is depicted in the *bc* plane and the *R* factor was 20.86% after solving using the SIR2014 software. The red, blue, gray, and white atoms denote oxygen, nitrogen, carbon, and hydrogen atoms, respectively. Four possible molecular structures of L-histidine (LH) are exhibited in **b** LH1, **c** LH2, **d** LH3, and **e** LH4 along with the atomic numbering scheme. Four models are considered based on the chemical formula ($C_6H_9N_3O_2$). **f** Variation of the *R* factor (%) after each refinement step as determined using the SHELXL software. **g** 2D $^1H$–$^{13}C$ and **h** $^1H$–$^{15}N$ invCP-VC spectra. The number of scans was 176 for $^1H$–$^{13}C$ spectrum and 112 for $^1H$–$^{15}N$ spectrum. The $^1H$–$^{13}C$ dipolar couplings obtained at $^1H$ chemical shift of 3.87 and 5.25 ppm were 15.19 and 15.40 kHz, respectively. The $^1H$–$^{15}N$ dipolar coupling obtained at $^1H$ chemical shift of 14.31 ppm was 7.11 kHz

**Table 1 RMSD values of chemical shifts for L-histidine[a]**

|      | $^1H$ (ppm) | $^{13}C$ (ppm) | $^{15}N$ (ppm) | Energy (Ry) |
|------|------------|---------------|---------------|-------------|
| LH1  | 0.63       | 3.29          | 6.39          | − 810.855   |
| LH2  | 2.87       | 7.68          | 15.44         | − 810.648   |
| LH3  | 2.88       | 4.06          | 8.23          | − 810.753   |
| LH4  | 2.93       | 9.39          | 5.69          | − 810.772   |

[a]The RMSD values were obtained between the SSNMR experimental and GIPAW-calculated isotropic chemical shifts. All the chemical shifts are listed in Supplementary Table 5

fail to identify the actual structure, NMR isotropic shifts are known to be reliable parameters for validating structures[6]. The optimization was performed using fixed cell parameters, and all of the atomic positions were allowed to relax. The calculations revealed that the LH1 structure was the most energetically favorable structure (Table 1). We also evaluated the refined structures using the root-mean-square deviation (RMSD) between the calculated ($\delta_{iso\_cal}$) and experimental ($\delta_{iso\_exp}$) isotropic chemical shifts for $^1H$, $^{13}C$, and $^{15}N$ (Table 1). The $^1H$, $^{13}C$, and $^{15}N$ isotropic shifts were measured via $^1H$–$^{13}C(^{15}N)$ heteronuclear correlation (HETCOR) experiments and assigned to the structures (see Supplementary Figures 2 and 3 and Supplementary Table 5). The LH2 structure exhibited the highest RMSD value for the $^{15}N$ isotropic chemical shifts, thereby allowing it to be safely excluded from the list of possible structures. Interestingly, the $^{13}C$ shifts were sensitive to the protonation states of N2 and N3. As the LH2 and LH4 structures exhibited higher RMSD values for the $^{13}C$ chemical shifts than

LH1 and LH3, the correct structure was considered not likely to involve protonation of N2, i.e., LH2 and LH4. Finally, comparison of the RMSD values for the $^1H$ chemical shifts completely ruled out the LH2, LH3, and LH4 structures and corroborated LH1 as being the correct structure. This is in agreement with the energies calculated using GIPAW as mentioned above. These findings are a straightforward consequence of $^1H$ chemical shifts being the most sensitive measure, as the differences in the structures of LH1 to LH4 originate from the hydrogen atom positions as shown in Fig. 2b–e. It is interesting to note that GIPAW optimization did not significantly affect the structure of LH1, whereas the other candidate structures underwent marked changes upon GIPAW relaxation (see Supplementary Figure 4). This result indicated that LH1 was the most energetically favorable structure from the beginning, although all of the structures were equally possible according to the *R* factors obtained from ED analysis. We next performed a set of GIPAW calculations on LH1 at various pseudopotentials to understand the statistical variation of bond lengths. The N3–H3 bond length was not largely dependent on the pseudopotential and the precision of the quantum computation was on the order of 0.01 Å (see Supplementary Figure 5). To evaluate the SSNMR-derived bond lengths, we further calculated the chemical shifts and energies as functions of the N3–H3, C2–H2, C5–H5, and C6–H6 bond lengths (see Supplementary Figure 6). In all cases, the bond lengths at the energy minima were 0.02–0.03 Å shorter than the SSNMR-derived bond lengths. This is consistent with the well-known relationship that SSNMR bond length based on the size of internuclear dipolar interactions are several percent longer than the distance averaged from atomic positions owing to the

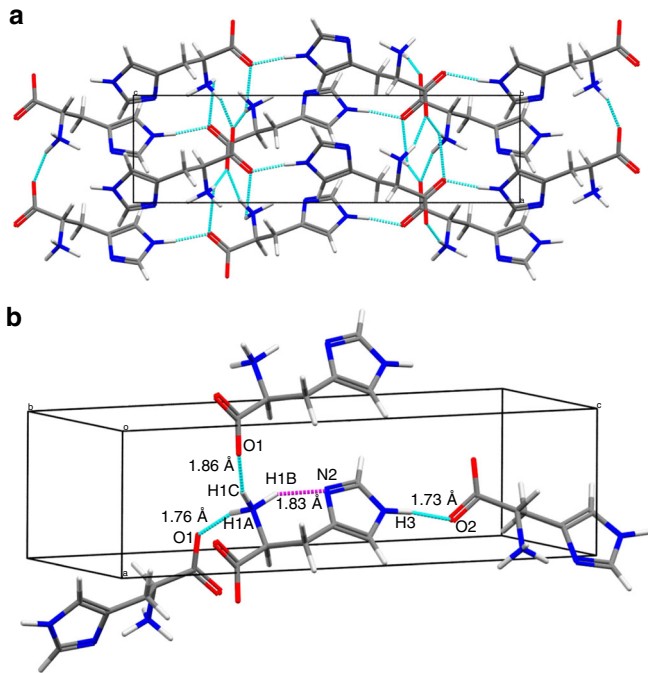

**Fig. 3** Hydrogen-bonding network in LH1. **a** Molecular packing structure of LH1 in the *ac* plane. **b** Extraction of hydrogen bonds showing the corresponding internuclear distances for each bond. The sky-blue and magenta broken lines indicate inter- and intramolecular hydrogen bonds, respectively. The structures were optimized via GIPAW calculations

influence of dynamical averaging[35,36]. This suggests that the SSNMR-derived bond lengths are reasonably accurate. As the calculated $^1$H–$^{13}$C($^{15}$N) bond lengths and isotropic $^1$H chemical shifts of LH1 are in agreement with the measured values, the bond lengths well reflected the hydrogen-bonding strength in the final structure. In addition, the molecular structure and lattice parameters of LH1 are almost identical to those previously determined via SC ND[37] (see Supplementary Table 2 and Supplementary Figure 7).

The molecular packing of the LH1 structure verified by ED, SSNMR, and calculation is described by the intermolecular hydrogen-bonding network (Fig. 3a). Figure 3b depicts the simplified hydrogen-bonding network between adjacent molecules involving four hydrogen bonds. One of the hydrogen atoms of the protonated amino group forms an intramolecular $NH_3^+$···N2 bond. The other two hydrogen atoms of the $NH_3^+$ group and the H3 hydrogen of the imidazole moiety are involved in intermolecular bonding of the N–H···O type. In contrast, the LH2 model with protonation at the N2 position cannot engage in intermolecular hydrogen bonding of N3–H3···O2 (see Supplementary Figure 8). In the case of LH3 and LH4, the 180° ring flip prevents the intramolecular hydrogen bonding of $NH_3^+$···N2. This indicates that the molecular packing is significantly influenced by hydrogen bonding, and therefore the hydrogen positions are critical for determining the crystal structure.

**Application of the combination of ED, SSNMR, and GIPAW.** As described in the previous subsection, the combined approach of ED, SSNMR, and GIPAW calculations permitted determination of the structure of L-histidine microcrystals. Next, this approach was applied to a previously unknown structure of the pharmaceutical cimetidine, which is a well-known histamine H2 receptor antagonist[38]. Cimetidine exhibits polymorphic behavior that includes crystal forms A, B, C, and D (or Z) and the

monohydrates M1, M2, and M3[39,40]. Although the structures of forms A, C, D, and M1 have been solved using SCXRD[39], to the best of our knowledge, the structure of form B remains unsolved. The main reasons for this are that form B crystals possess a needle shape, rendering the growth of SCXRD-quality crystals challenging[41], and are difficult to prepare in high purity. Although we have been able to successfully and carefully prepare form B, contamination with form C is always observed. The presence of this mixture of crystal forms limits the application of PXRD for structure determination. Although $^{13}$C SSNMR spectra have been used to distinguish different crystal forms, forms B and C exhibit very similar chemical shifts and there are no distinct $^{13}$C chemical shifts corresponding to form C (Fig. 4a, b)[42,43]. However, the $^1$H–$^{15}$N SSNMR spectra (Fig. 4c, d) contained separate peaks for each form, where the peaks indicated with red asterisks correspond to form C. The peaks corresponding to each form were assigned by comparing the relative signal intensities for samples prepared under different conditions. The $^{15}$N, $^{13}$C, and $^1$H chemical shifts were assigned using 2D $^1$H–$^{15}$N/$^{13}$C spectra (the atomic numbering scheme is shown in Fig. 5b), and the $^{13}$C chemical shifts shown in Fig. 4a, b are in good agreement with those reported in the literature[42,43]. Considering the chemical formula ($C_{10}H_{16}N_6S$), the $^{13}$C and $^{15}$N spectra contained additional peaks that displayed splitting, such as C1/C11, C5/C15, C6/C16, N3/N9, N4/N10, and N6/N12. These peak splittings might be attributed to the existence of two magnetically inequivalent sites for the chemically equivalent atoms of form B. The three pairs of NH correlations observed in the 2D $^1$H–$^{15}$N spectra suggest that only one of the imidazole nitrogen atoms was protonated, which is analogous to the case of L-histidine. Using the same method as that used to determine the structure of L-histidine, the initial crystal structure was solved using several sets of ED patterns (see Supplementary Figure 9). Analysis of the individual data sets using XDS revealed the lattice parameters and the major structures in agreement were selected for the next step. Three data sets were merged to obtain the final structural solution (see Supplementary Table 6). The CB structure belonged to space group $C2/c$ and possessed monoclinic crystal symmetry with lattice parameters of $a = 55.45(15)$ Å, $b = 5.0$ Å, $c = 18.72(6)$ Å, and $\beta = 100.4(5)°$ (see Supplementary Table 7). The crystal structure after phasing in the SIR2014 software included two conformations as depicted in Fig. 5a–d; one of the molecules adopts an extended conformation (I) and the other possesses a bent conformation (II). This supports the assertion that the peak splitting in the $^1$H–$^{15}$N/$^{13}$C SSNMR spectra was attributable to the magnetically inequivalent sites of two distinct conformations. The R factor was 20.02% for the initial structure. As some carbon, nitrogen, sulfur, and hydrogen atoms were misassigned (see Supplementary Figure 10), the atomic positions of the non-hydrogen elements were corrected and the hydrogen atoms were removed using SHELXL. The removal of hydrogen considerably increased the R factor to 23.74% (see Fig. 6a and Supplementary Table 8). In contrast to the case of L-histidine, there was no ambiguity in the orientation of the imidazole ring of cimetidine. However, it was still unclear which of the imidazole nitrogen atoms was protonated. In the difference potential map, most of the hydrogen atoms could not be placed automatically (see Supplementary Figure 11). Although the difference potential map revealed several possible hydrogen atom positions, unreasonable bond lengths or bond angles prevented further refinement. Based on the two conformations, four candidate models for CB were considered as depicted in Fig. 5a–d. The addition of hydrogen atoms to these four candidates improved the R factors to 19.79–20.26% (Fig. 6a). The hydrogen positions were then refined using the internuclear distances measured via SSNMR: C–H distance = 1.12 Å (± 0.01

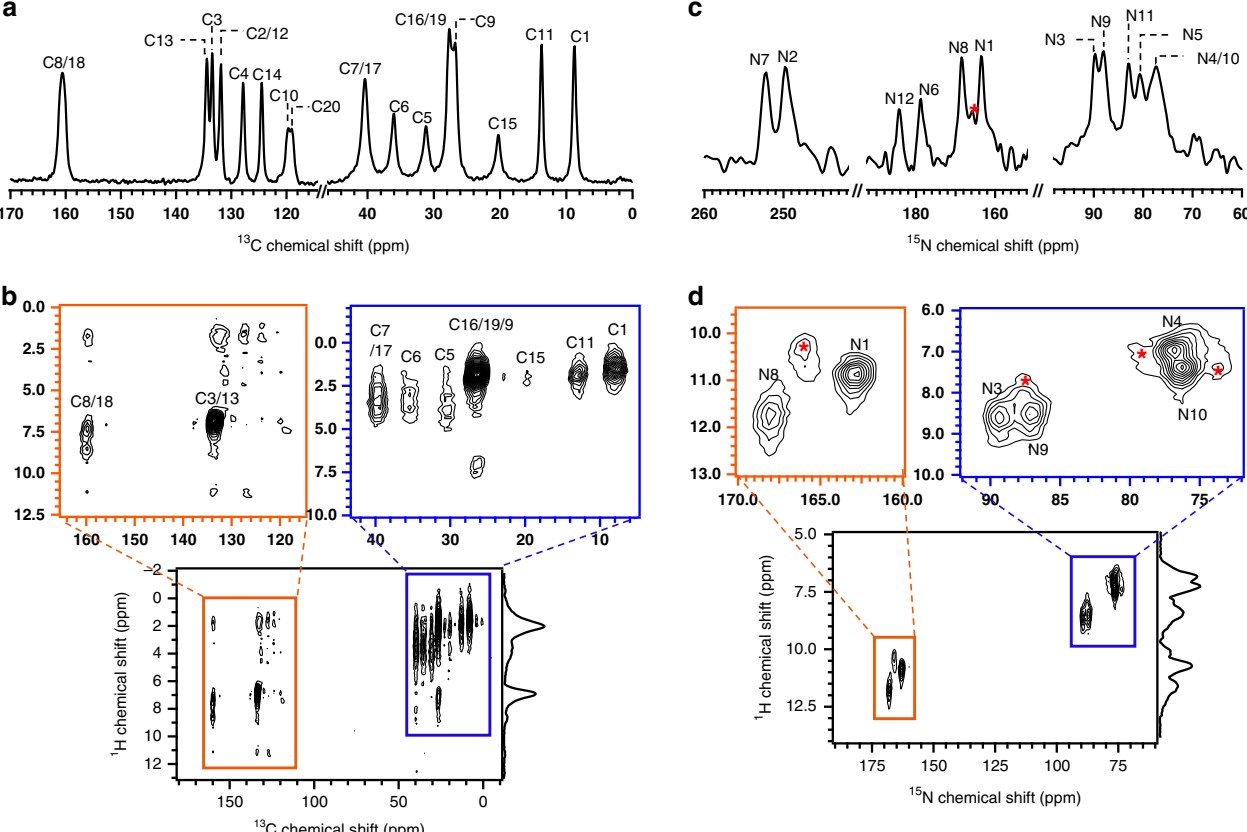

**Fig. 4** Structural assignment of cimetidine form B using SSNMR spectra. The SSNMR spectra were measured for a mixture of forms B and C. **a** 1D $^{13}$C and **c** $^{15}$N CPMAS spectra measured using 500 and 5000 scans, respectively. The CP contact time was adjusted to 2 ms for $^{13}$C and 1 ms for $^{15}$N. Only the spectral range of interest is displayed. **b** 2D $^1$H–$^{13}$C and **d** $^1$H–$^{15}$N HETCOR spectra over the full spectral range with expansions of the regions of interest as indicated by the blue and orange boxes. The atom labels are identical to those shown for CB2 in Fig. 5b. The contact time was 2 ms during the first CP transfer and 0.5 ms during the second CP transfer for both spectra. The 2D $^1$H–$^{13}$C spectrum was acquired using 40 scans and 128 rows in the indirect dimension and the 2D $^1$H–$^{15}$N spectrum was acquired using 24 scans and 64 rows. The $^1$H–$^{15}$N correlation peaks from form C are indicated with red asterisks

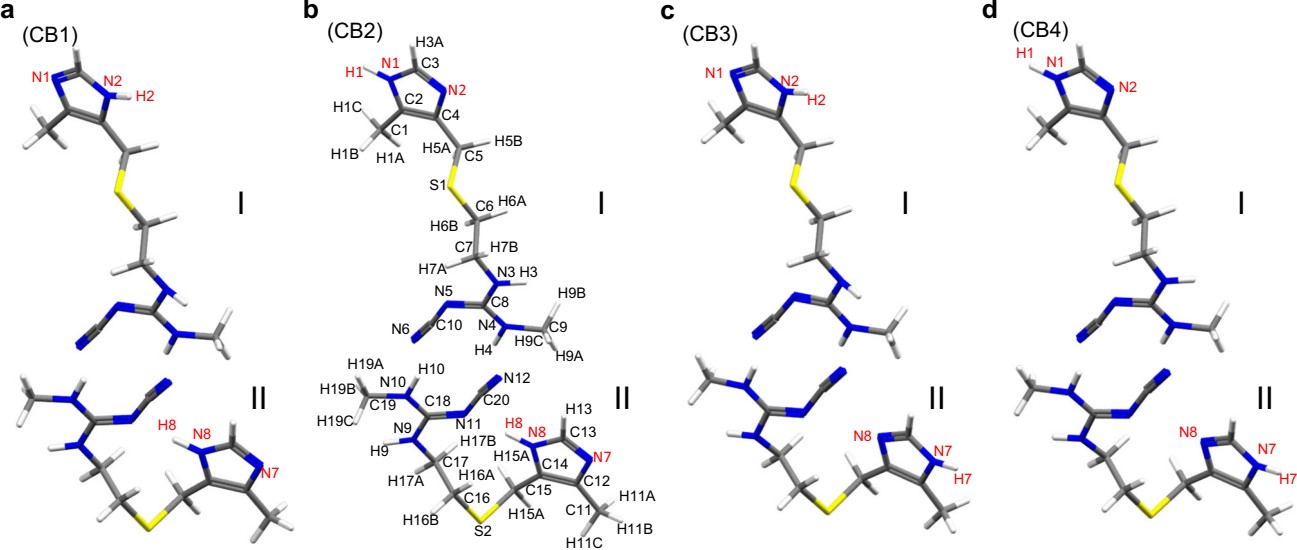

**Fig. 5** Four possible molecular structures of cimetidine form B. The cimetidine form B (CB) structures of **a** CB1, **b** CB2, **c** CB3, and **d** CB4 depend on the protonation site of the imidazole ring. The atomic numbering scheme is shown. Each structure is composed of two conformations; I and II refers to an extended and a bent conformation, respectively. The yellow, blue, gray, and white atoms denote sulfur, nitrogen, carbon, and hydrogen atoms, respectively

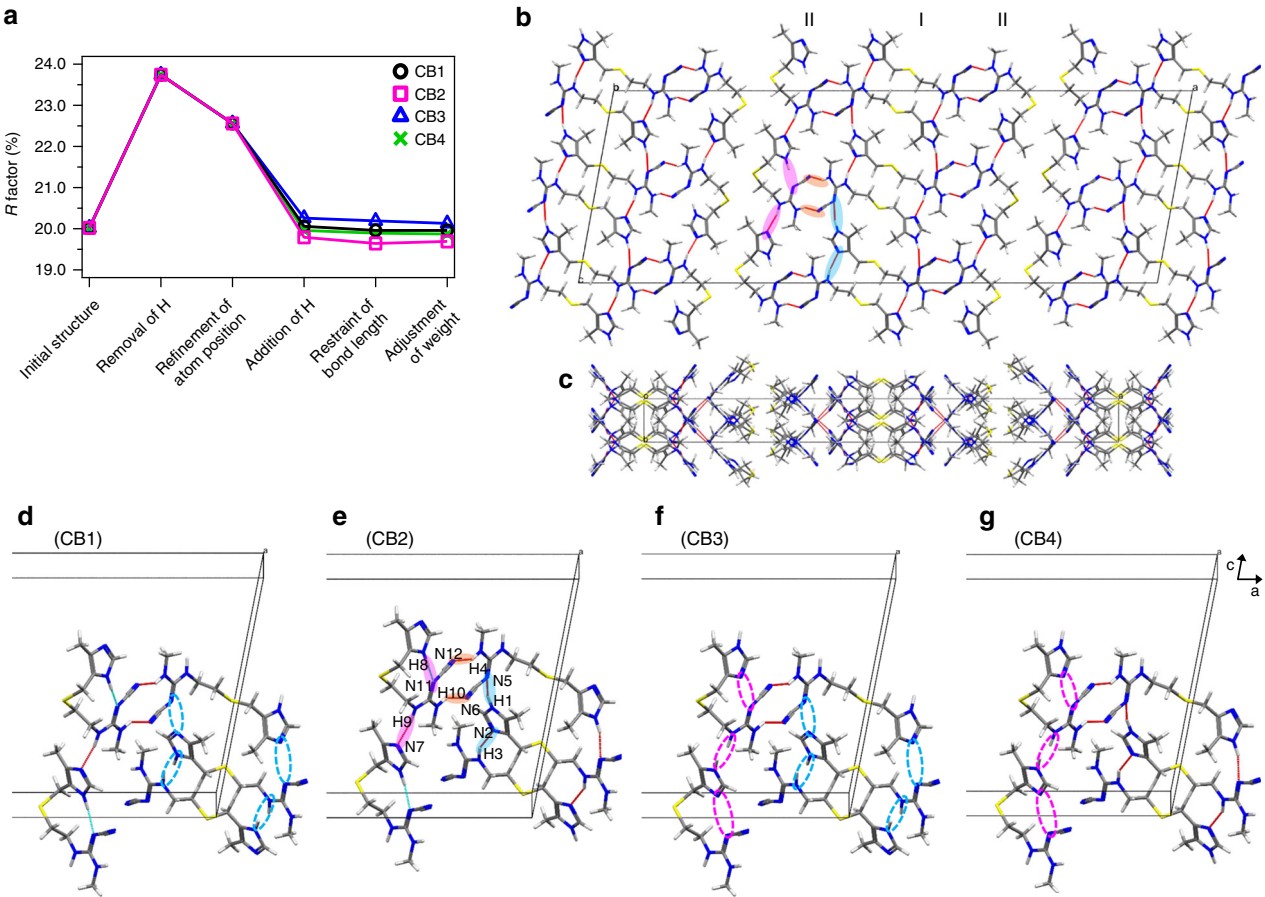

**Fig. 6** Cimetidine form B crystal structure and its hydrogen-bonding network. **a** Variation of the *R* factor (%) after each refinement step as determined using the SHELXL software. Hydrogen-bonding network of the CB2 structure in **b** *ac* and **c** *ab* planes. For the sake of simplicity, only intermolecular hydrogen bonds are shown as red lines. **b**, **e** The pink and sky-blue solid circles indicate the hydrogen bonds between I conformations and between II conformations, respectively. The orange solid circles represents the hydrogen bonds between I and II conformations. **d**–**g** Extraction of hydrogen bonds for the **d** CB1, **e** CB2, **f** CB3, and **g** CB4 structures. The unit cell consists of adjacent molecules connected by intermolecular hydrogen bonds. The inter- and intramolecular hydrogen bonds are depicted by red and sky-blue lines, respectively. The magenta and sky-blue broken circles denote the missing hydrogen bonds of H8⋯N11 and H9⋯N7, and H1⋯N5 and H3⋯N2, respectively. In CB2, the lengths of the hydrogen bonds were as follows: N1–H1⋯N5 = 2.172 Å, N3–H3⋯N2 = 1.996 Å, N8–H8⋯N11 = 1.981 Å, N4–H4⋯N12 = 1.927 Å, N9–H9⋯N7 = 1.920 Å, and N10–H10⋯N6 = 1.872 Å. All the structures **b**–**g** were optimized via GIPAW calculations

Å) for the imidazole ring and 1.06 Å (± 0.01 Å) for C5–H5, C6–H6, C7–H7, C15–H15, C16–H16, and C17–H17, and N–H distance = 1.06 Å (± 0.04 Å) for the imidazole ring and 1.04 Å (± 0.06 Å) for the guanidine group (see Supplementary Figure 12). The $^{1}$H–X bond lengths are presented in Supplementary Table 9. Restraining the bond lengths barely improved the *R* factors (Fig. 6a), and weight adjustment afforded similar final *R* factors of 19.96%, 19.69%, 20.13%, and 19.87% for the CB1, CB2, CB3, and CB4 models, respectively, which did not permit determination of the correct structure. Thus, we performed GIPAW optimization of these models. The calculated energies revealed that CB2 was the most energetically favorable structure (Table 2). The final validation was conducted by evaluating the RMSD values for the $^{1}$H, $^{13}$C, and $^{15}$N chemical shifts, all of which indicated that CB2 was the correct structure (Table 2 and see the details in Supplementary Table 10, and Supplementary Figure 13). As described above for L-histidine, the CB2 structure underwent the smallest change upon GIPAW relaxation, further confirming the veracity of this structure (see Supplementary Figure 14). We further evaluated the SSNMR-derived bond lengths by calculating the chemical shifts and energies as functions of the N1–H1, N3–H3,

**Table 2 RMSD values of chemical shifts for cimetidine form B[a]**

|  | $^{1}$H (ppm) | $^{13}$C (ppm) | $^{15}$N (ppm) | Energy (Ry) |
|---|---|---|---|---|
| CB1 | 1.23 | 4.50 | 12.77 | − 2192.289 |
| CB2 | 0.73 | 3.35 | 3.96 | − 2192.543 |
| CB3 | 1.61 | 6.28 | 17.04 | − 2192.069 |
| CB4 | 1.21 | 5.68 | 13.31 | − 2192.308 |

[a]The RMSD values were obtained between the SSNMR experimental isotropic and GIPAW-calculated isotropic chemical shifts. All the chemical shifts are listed in Supplementary Table 10

N4–H4, N8–H8, N9–H9, and N10–H10 bond lengths (see Supplementary Figure 15). All the N–H bond lengths were 0.01–0.03 Å shorter than the SSNMR-derived bond lengths owing to the effect of dynamical averaging as described for L-histidine. The consistency of the bond lengths obtained from the GIPAW calculations and SSNMR experiment indicated that the SSNMR-derived bond lengths were reasonably accurate. Fortunately, we were also able to prepare large and high-quality single crystals of CB and determine their crystal structure using SCXRD. This

structure was almost identical to the CB2 structure, providing independent validation of the ED/SSNMR results (see Supplementary Figure 16 and Supplementary Table 7). It is interesting to compare the intramolecular $^{13}C-^{13}C$ distances between C3 (C13) and C9 (C19). The C3–C9 and C13–C19 distances in the extended and bent conformations were 11.044 Å and 7.179 Å, respectively. In a previous NMR study of cimetidine, the rotational resonance curves with selective double-$^{13}C$ labeling indicated a distance of 5.5 Å between C3 (C13) and C9 (C19)[44], which differs considerably from the values obtained from the structure solved using ED. This apparent discrepancy is because these curves were dominated by the intermolecular distance rather than the intramolecular distance and, taking this into account, the previous NMR results are consistent with the current structure. The details are described in the Supplementary Note 1.

The CB2 structure can be explained by the inter- and intramolecular hydrogen-bonding networks depicted in Fig. 6b–g. The molecules in conformation I are arranged in a head-to-tail manner owing to the intermolecular hydrogen bonds of N1–H1···N5 and N3–H3···N2 between the imidazole ring and the guanidine group, as indicated by the sky-blue solid circles, whereas the molecules in conformation II are stabilized by the intermolecular hydrogen bonds of N9–H9···N7 and the intramolecular hydrogen bonds of N8–H8···N11, as indicated by the magenta solid circles (Fig. 6b, e). Furthermore, the layers composed of molecules in conformations I and II are alternately arranged in the order II–I–II along the $a$ axis by the hydrogen bonds of N4–H4···N12 and N10–H10···N6, as indicated by the orange solid circles (Fig. 6b, e). However, the protonation of N7 in CB3/4 renders conformation II unable to engage in intra- or intermolecular hydrogen bonding, whereas the protonation of N2 in CB1/3 disrupts the intermolecular hydrogen bonding of conformation I (broken circles in Fig. 6d–g). This clearly indicates that the formation of the hydrogen-bonding network depends on the protonation position of the imidazole ring. In fact, conformations I and II of form B are similar to the molecular structures reported for form C[40] and form A[45], respectively. However, in the structure of form A the N7 of the imidazole ring is protonated and the cyanoguanidine group is rotated by 180°, which induces intramolecular hydrogen bonding between the imidazole ring and cyanoguanidine group (N8···H10–N10). Therefore, the positions of the hydrogen atoms exert a crucial influence on the conformation and crystal packing.

## Discussion

The structures of molecular crystals, including the hydrogen atom positions, were successfully determined from single crystals smaller than 1 μm using the combination of ED, SSNMR, and first-principles quantum calculations. The utility of the combined approach was first demonstrated for the known structure of orthorhombic L-histidine. The results demonstrated the importance of accurately determining both the structure and the hydrogen positions for obtaining a comprehensive understanding of the hydrogen-bonding network in the organic crystals. Although ED using the rotation method allowed localization of most of the atoms, the positions of the hydrogen atoms remained uncertain based solely on the ED patterns, and the carbon, nitrogen, and oxygen atoms were difficult to distinguish if dynamic scattering was not considered during the analysis. This led to multiple candidate structures. Internuclear distance measurements and isotropic chemical shifts with the help of GIPAW calculations were used to accurately determine the hydrogen atom positions and unambiguously assign the carbon, nitrogen, and oxygen atoms. By comparing the experimental and calculated NMR chemical shifts, GIPAW calculations combined with ED

and SSNMR results permitted determination of the correct structure. This approach was next applied to CB, allowing the crystal structure to be determined for the first time as monoclinic with space group $C2/c$ and lattice parameters of $a = 55.45(15)$ Å, $b = 5.0$ Å, $c = 18.72(6)$ Å, and $\beta = 100.4(5)°$. This model was successfully refined to afford a final $R$ factor of 19.69%. Several sets of ED data, obtained from three to five single crystals, were sufficient for complete determination of the 3D crystal structure. As the samples contained a mixture of crystal forms, the rigorous preparation of pure samples was not necessary. Thus, the combined approach reported herein can be applied to numerous nano- to microcrystalline samples, including pharmaceutical formulations, metal–organic frameworks, and other compounds.

## Methods

**Sample preparation.** L-histidine powder ($C_6H_9N_3O_2$) (Wako Pure Chemical Industries Ltd., Japan) was dissolved in distilled water at 343.15 K, and the resulting solution was maintained at 323.15 K for 2 days followed by filtration to afford the recrystallized orthorhombic structure[46]. The recrystallized microcrystals was dried at 313.15 K and identified as orthorhombic using PXRD. CB ($N$-cyano-$N'$-methyl-$N''$-[2-[(5-methyl-1$H$-imidazol-4-yl)methylthio]ethyl]guanidine(I)) ($C_{10}H_{16}N_6S$) was crystallized by slowly cooling a hot 15% (w/w) aqueous solution of cimetidine at 343.15 K[47]. Sharp needle-shaped crystals were obtained in good purity with a maximum length of ~ 40 μm.

**Solid-state NMR measurements.** Solid-state NMR measurements of the microcrystalline powder samples of orthorhombic L-histidine and CB were conducted at 14.01 T using a JNM-ECZ600R spectrometer (JEOL RESONANCE Inc., Japan) operating at a $^1H$ resonance frequency of 599.7 MHz. $^1H-^{13}C/^{15}N$ cross-polarization magic-angle spinning experiments were performed using a 3.2 mm double-resonance MAS probe (JEOL) operating at 20 kHz MAS. The $^1H$ radio frequency (rf) field strength was 100.0 kHz at a $\pi/2$ pulse length of 2.5 μs. Two-pulse phase modulation was used for proton decoupling during acquisition. 2D $^1H-^{13}C/^{15}N$ HETCOR and 2D $^1H-^{13}C/^{15}N$ invCP-VC spectra were acquired using a 1 mm double-resonance MAS probe (JEOL) operating at 70 kHz MAS. The $\pi/2$ pulse length was 1.15 μs for $^1H$, 1.1 μs for $^{13}C$, and 1.9 μs for $^{15}N$. The 2D HETCOR spectra were collected via $^1H$-detected $^1H-X$ CP heteronuclear single quantum coherence (HSQC)[48]. In all of the 2D $^1H-X$ HETCOR experiments, ramped CP was used for magnetization transfer between $^1H$ and X under Hartmann–Hahn matching conditions. Heteronuclear decoupling was accomplished using the wideband alternating-phase low-power technique for zero residual splitting (WALTZ) pulse sequence for both the indirect and direct dimensions and it was set to 11 kHz for $^{13}C$ and 9 kHz for $^1H$. During the homonuclear rotary resonance-recoupling period, the unwanted $^1H$ magnetization was removed prior to $^1H$ observation. The invCP-VC spectra were measured using the same NMR sequence as for the $^1H-X$ CP-HSQC experiments except for the second CP conditions[31]; the second CP for X → $^1H$ CP transfer was carried out as a function of variable contact time with a constant rf field strength. The relaxation delays for cimetidine and L-histidine were 15 and 6 s, respectively. All spectra were processed using the Delta software (JEOL). The bond lengths were determined using Equation (1). The peak separation ($\Delta$) in the indirect dimension reflects the $^1H-X$ dipolar coupling, which was converted to the $^1H-X$ distance using the following relationship:

$$d_{1H-X}(\text{Å}) = \left( \frac{120.1}{\sqrt{2}\Delta(\text{kHz})} \frac{\gamma_X}{\gamma_{1H}} \right)^{1/3} \tag{1}$$

where $\gamma_{1H}$ and $\gamma_X$ are the gyromagnetic ratios of $^1H$ and X, respectively, and $\sqrt{2}$ is the scaling factor[49].

**ED measurements.** The ED patterns of the L-histidine and cimetidine crystals were measured using a JEM-2200FS transmission electron microscope (JEOL Ltd., Japan) operating at 200 kV with continuous rotation of the sample. To minimize the electron radiation damage, all of the measurements were performed at a low dose rate of 8 e$^-$ nm$^{-2}$ s$^{-1}$ using a cryogenic sample holder (Gatan 914, Gatan Inc., US) maintained at a temperature of 96.15 K. The diffraction data were recorded using a high-sensitivity CCD camera (Gatan Ultrascan, Gatan Inc., US) with ×2 binning (1024 × 1024 pixels). The camera length (603.2 mm) was calibrated using a gold polycrystal specimen as a standard. The low temperature and electron dose helped to afford undamaged diffraction patterns. To avoid structural changes owing to the electron beam irradiation during the measurement, several data sets were measured for several different crystals. A typical rotation series for one set of diffraction patterns contained ~ 40 frames, which were collected using holder rotation steps of ~ 1.18° and covered a range of ~ 40° over 4 min. The granular particles of L-histidine and cimetidine crystals were ground in a mortar and distributed on a ultra-thin carbon film to obtain the TEM specimens. The diffraction patterns were recorded for crystals of submicrometer size (200 nm to 2 μm).

**Density functional theory calculations**. The geometrical optimization and chemical shift calculations were conducted using the density functional theory programs pw and qe-gipaw in the Quantum ESPRESSO package version 6.1. The cif2cell program was used to convert atomic coordinates from CIF to the unit cell. The unit cell contained 80 atoms for L-histidine and 264 atoms for cimetidine B. The Monkhorst–Pack $k$-point grid was generated using a resolution of 0.4 Å$^{-1}$. We observed a minor ($\pm 0.01$ Å) dependence on the pseudopotential and used X.pbe-trm-new-gipaw-dc.UPF as the pseudopotential for the atom X[50]. The kinetic energy cutoff for the wavefunctions (ecutwfc) was set to 80 Ry. The coordinates of all atoms were optimized with a fixed cell. NMR chemical shifts were obtained by subtracting the calculated chemical shifts from the reference shifts, which were adjusted to give the best agreement with the experimental data.

**SCXRD measurements**. A colorless needle-shaped single crystal of cimetidine with dimensions of $0.155 \times 0.045 \times 0.034$ mm was selected for the SCXRD measurements. The specimen was fixed on a glass capillary and mounted on a Rigaku AFC-8 diffractometer equipped with a Saturn 70 CCD detector. The diffraction data were collected using the oscillation method with a rotational angle of 0.5° and MoK$\alpha$ radiation ($\lambda = 0.71073$ Å) at 90 K. The diffraction data were collected up to 50.152° in $2\theta$. The cell constants were $a = 54.938(3)$ Å, $b = 4.8958(3)$ Å, $c = 18.5234(13)$ Å, $\beta = 100.297(4)$°, $V = 4901.9(5)$ Å$^3$, and $Z = 16$, and the space group was $C2/c$. A total of 37293 reflections were measured and merged. Finally, 4320 independent reflections were obtained with $R_{int} = 0.1055$. The structure was solved via the direct method using the SIR92 software[51] and refined via the least-squares method using the SHELXL-2018/1 software[52]. After refinement of the non-hydrogen atoms with anisotropic temperature factors, all of the hydrogen atoms were located on Fourier maps. The hydrogen atoms were refined by applying riding models. The final values of $R(F)$, $wR(F2)$, and $S$ were 0.0885, 0.2259, and 1.144 for the 3195 reflections with $I > 2\sigma(I)$.

## Data availability

All data needed to evaluate the conclusions in the paper are present in the paper and/or the Supplementary Materials. Data that support the findings of this study have been deposited to the Cambridge Structural Database (CSD) and are accessible through CCDC numbers of 1889705 (ED/SSNMR/GIPAW) for L-histidine, and 1889706 (ED/SSNMR/GIPAW) and 1879225 (SCXRD) for cimetidine. All other relevant data are available from the corresponding author on request.

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

## Acknowledgements

We acknowledge Dr. Daisuke Hashizume of RIKEN-CEMS for the SCXRD measurements and analysis. This work was partially supported by a Japan Society for the Promotion of Science Grant-in-Aid for Scientific Research 16H04757 (to K.Y.), a Japan Society for the Promotion of Science Grant-in-Aid for Challenging Exploratory Research 24657111 (to K.Y.), and the Japan Science and Technology Agency SENTAN program (to K.Y.).

## Author contributions

Y.N. conceived and designed the project. K.Y. supervised the data collection and processing of ED patterns, and C.G.A., Y-l.H. and K.Y. analyzed the ED data. H.I., A.S., T.F., Y.A., S.M., T.O. and Y.N. acquired the ED data. C.G.A. and Y-l.H. analyzed the experimental ED and NMR data. H.C. prepared the samples and collected the NMR data. T.Y. performed the GIPAW calculations.

## Additional information

**Competing interests:** The authors declare that they have no competing interests.

