## [Peer Review File · Nature Communications]

Reviewers' comments:

Reviewer #1 (Remarks to the Author):

In this article, the authors demonstrated how to successfully solve the structure of molecular crystals together with hydrogen bonding network by combining the ED and SSNMR with principle quantum computation. The manuscript deals with an important topic which is of strong interest in many different fields and it is of particular interest for all those cases in which X-ray diffraction fails, as for nano-sized crystals, and for samples which are characterized by different components. The consecutive steps that lead from multiple candidates to the correct structure have been clearly and logically explained, highlighting the main contribution of each employed technique. The approach is novel, the work has been carefully carried out and the evidences provided are convincing.

I would recommend the publication of this manuscript in Nature Communication with several minor comments.

1. The text must be revised since there are many typos and grammar errors.

See for example:

- at line 31: "is" should be changed with "are"

- at line 50: "cannot be solved by both SCXRD and SCND" should be changed with "cannot be solved by either SCXRD or SCND"

- at line 52: "framework" with "frameworks"

- at line 63: "may disturb analysis" with "may disturb the analysis"

- at line 90: "salt and cocrystal" with "salts and cocrystals"

- at line 127: "positions of hydrogen" with "positions of hydrogen atoms"

- at line 128: "from in Coulomb potential map" with "from Coulomb potential maps"

- at line 138: "this molecule is zwitterions" with "this molecule is a zwitterion"

- at line 156: "bond length" with "bond lengths"

-at line 157: "it does not take into account from factors of length variation" with "they do not take factors of length variation into account"

-etc.

2. At line 168: no evaluation of the N-H distance on NH₃⁺ is reported. Please explain.

3. At line 239: "there is little difference in ¹³C chemical shift between form B and C as shown in Fig. 4a, b". Where is form C in the reported spectra?

4. At line 245-247 "In the ^{13}C and ^{15}N spectra, two separate peaks from the chemically equivalent site of form B are attributed to the presence of two magnetically inequivalent molecules in the unit cell. " No clear indication has been provided on the two separate peaks considered in the ^{13}C and ^{15}N spectra.

5. At line 296 the authors mention the magenta and sky-blue circles in figure 6b, which are not present. Do the authors refer to figure 6c? It should be clarified or corrected.

6. In Figure 6c, "I" and "II" labels would help to understand the description of the molecular arrangement. Also, the numbering of the N and H atoms involved in the bonds described in the main text should be added to facilitate the reader.

Reviewer #2 (Remarks to the Author):

This is a nice paper reporting on the combined use of solid-state NMR, electron diffraction, and quantum mechanical calculations of chemical shifts, to determine crystal structures of small molecules. This approach has been dubbed "NMR crystallography" and has become popular in recent years for elucidation of crystal structures, as evidenced by over 200 papers published on the subject in the recent decade or so. The main advantage of combining the different techniques (most commonly, X-ray crystallography, solid-state NMR, and QM calculations) is that proton positions and protonation states can be determined, which are typically not visible in most of the X-ray or ED structures. Hence significant effort has been put into the development of this approach by multiple NMR groups across the globe.

The current paper represents yet another example of such an approach, using a well-studied example of L-histidine whose crystal structures are well known and a cimetidine form B with unknown structure. The authors demonstrate that their approach is viable, using L-histidine, where they delineate the only structure consistent with all of the data. They then apply this strategy to cimetidine form B.

The main novelty of the work presented by the authors is that, instead of X-ray crystallography, they employ electron diffraction. The rest of the study follows the large body of the previously published literature. The technical quality is overall high. However, all of the concepts, the experimental and computational methods are well established and straightforward. The solid-state NMR data, while of

high quality, reproduce the multiple reports from many different groups. There is no new information - either technical or conceptual there.

In this reviewer's opinion, this work is valuable for researchers interested in small-molecule crystal structures, including those of pharmaceutical solids. Since there are no major methodological innovations, the work would be much better suited for publication in a more specialized journal, such as Journal of Physical Chemistry, PCCP or PCP. The authors should also fix multiple grammatical errors and typos in the revised manuscript.

Reviewer #3 (Remarks to the Author):

The manuscript describes the structure determination of two organic pharmaceutical compounds with the emphasis on the correct determination of the hydrogen bonding network and correct assignment of potentially ambiguous atomic types. The method used is the combination of 3D electron diffraction with ssNMR and DFT calculations. It is convincingly shown that ssNMR combined with simulations provides an efficient tool to supplement information, which is difficult to obtain solely by 3D electron diffraction.

The novelty of the manuscript is not in the use of ssNMR to complete diffraction data. This is being frequently done, especially for pharmaceuticals. The novelty may be seen in the fact that the same method that is common for x-ray diffraction may be successfully used in combination with electron diffraction to obtain accurate structures from micro- and nanocrystals. This fact as such is not really surprising, but it is certainly useful to prove it and to alert the scientific community about this option.

While the general outline of the manuscript is fine, the discussion sound and the results convincing, I got a general impression that, in order to strengthen the importance of the proposed method, the potential and ability of structure solution by electron diffraction is underestimated at several places of the manuscript, either on purpose or due to the lack of experience. I will describe the instances of this problem in more detail below, and I suggest this is corrected, otherwise a false impression of the potential of ED will be conveyed.

Another problem is that the manuscript would strongly benefit from proof reading focused on the correction of language and wording.

Specific comments:

General note: the word precision is used often in the manuscript, but possibly not always in the correct meaning. The word precision means "small statistical error". Often the more appropriate word is "accuracy", which means "deviation from the correct value".

line 47: "...with high precision." What is meant by "high precision? In general the hydrogen positions need not be known to very high accuracy to understand the hydrogen bonding network.

lines 88-91: the same problem as in previous comment. The accuracy of the hydrogen positions was about 0.1 Angstroms or better in the cited reference. This is sufficiently high accuracy to determine the hydrogen bonding network and to distinguish salt from cocrystal. It is not important to determine the hydrogen position to a high accuracy, but with high reliability, i.e. be sure that we see all hydrogens in their correct places. The cited work, but also other works (Clabbers et al., *Acta Crystallogr. A* 2019, 75, 82–93; Hynek, J. et al. *Angew. Chem. Int. Ed.* 2018, 57, 5016–5019.) show that this is quite well possible in many cases.

line 135: the difference between 22.71 and 23.86 seems relatively large. The claim that electron diffraction is unable to distinguish the two orientations is thus misleading. Moreover, the distinction can be also made by inspection of the displacement parameters and/or difference Fourier map. My suspicion is that these additional checks would further support the estimation made from the R-values, making the distinction between the two structures almost unambiguous. It would be more appropriate to say that the distinction may not be 100% certain and may need confirmation. However, the general claim that the distinction of atoms with close atomic numbers may be difficult, is correct, unless dynamical diffraction effects are taken into account, in which case ambiguities are very rare.

line 154: the supplementary figure 1 is used to support the claim that hydrogen positions cannot be found with sufficient precision. However, the figure contains the total potential map with relatively high isosurface level, but the hydrogen positions (and other weak features) are best found in a difference potential map (as e.g. in ref. 13). The claim on line 154 should be supported by a difference potential map showing no or only a few maxima corresponding to hydrogen positions.

line 167: please specify the standard uncertainty of the bond lengths. The high accuracy of the hydrogen structure determination by ssNMR is a central topic of the paper and the precision of the bond length determination should thus be carefully computed and given in the manuscript.

line 168: it appears to me that the authors are comparing the calculated bond lengths with the default values in SHELXL for x-ray diffraction data. If this is the case then it is not correct, because the default values for x-ray data are shorter than the real distances due to the shift of the hydrogen's electron to the covalent bond. It is not appropriate to use these values for electron diffraction. The comparison should be done with respect to some accepted source of tabulated X-H distances.

line 173: please give the values for R-values for all four structures. The range 19.81 - 21.76 appears relatively large to be classified as "very similar" - see my comment for line 135.

line 269: here again the difference potential map should be shown to support the inability of ED data to locate hydrogen atoms.

Supplementary table 9: please indicate standard uncertainties of the ssNMR bond lengths in the table.

Reviewers' comments:

Reviewer #1 (Remarks to the Author):

In this article, the authors demonstrated how to successfully solve the structure of molecular crystals together with hydrogen bonding network by combining the ED and SSNMR with principle quantum computation. The manuscript deals with an important topic which is of strong interest in many different fields and it is of particular interest for all those cases in which X-ray diffraction fails, as for nano-sized crystals, and for samples which are characterized by different components. The consecutive steps that lead from multiple candidates to the correct structure have been clearly and logically explained, highlighting the main contribution of each employed technique. The approach is novel, the work has been carefully carried out and the evidences provided are convincing.

I would recommend the publication of this manuscript in Nature Communication with several minor comments.

Our response:

We thank the reviewer for the positive comments.

1. The text must be revised since there are many typos and grammar errors.

See for example:

- at line 31: "is" should be changed with "are"
- at line 50: "cannot be solved by both SCXRD and SCND" should be changed with "cannot be solved by either SCXRD or SCND"
- at line 52: "framework" with "frameworks"
- at line 63: "may disturb analysis" with "may disturb the analysis"
- at line 90: "salt and cocrystal" with "salts and cocrystals"
- at line 127: "positions of hydrogen" with "positions of hydrogen atoms"
- at line 128: "from in Coulomb potential map" with "from Coulomb potential maps"
- at line 138: "this molecule is zwitterions" with "this molecule is a zwitterion"
- at line 156: "bond length" with "bond lengths"
- at line 157: "it does not take into account from factors of length variation" with "they do not take factors of length variation into account"
- etc.

Our response:

We have carefully corrected the typographical and grammatical errors, including those pointed out by the reviewer. In addition, the revised manuscript was submitted to a

proofreading service to further correct any remaining errors and improve the overall readability of the manuscript.

2. At line 168: no evaluation of the N-H distance on NH₃⁺ is reported. Please explain.

Our response:

We could not include an evaluation of the N–H distances (size of N–H dipolar interactions) for NH₃⁺ owing to the limitations of the solid-state NMR method used. The N–H dipolar interactions of the NH₃⁺ moiety were considerably smaller than those of the N–H moiety as they were scaled by a factor of $P2(\cos \theta) \approx -0.33$ owing to rapid rotation/jump around the C3 (C–N) axis, where θ represents the angle between C–N and N–H (Paluch, P., et al., J. Mag. Reson. 2013, 233, 56). Unfortunately, the inversely detected CP-VC method, which we used to determine the N–H dipolar couplings, does not provide an accurate size for small dipolar interactions. In fact, as CP-VC spectra always contain an intense center peak, the small dipolar splitting is obscured by partial overlap with the center peak. We have now clearly mentioned this in the revised manuscript.

On line 189:

“The N–H distances of the NH₃⁺ moiety were not evaluated owing to the technical difficulty of determining small dipolar couplings of NH₃⁺, which were dynamically scaled by a factor of approximately -0.33 ³².”

3. At line 239: “there is little difference in ¹³C chemical shift between form B and C as shown in Fig. 4a, b”. Where is form C in the reported spectra?

Our response:

The ¹³C spectra of forms B and C were reported in Pacilio, J. E., et al., J. Chem. Educ. 2014, 91, 1236 (ref. 43) and Middleton, D. A., et al., J. Pharm. Sci. 1997, 86, 1400 (ref. 42). As shown in these references, it is difficult to distinguish the two forms using ¹³C spectra because they exhibit very similar ¹³C chemical shifts (the difference is approximately 0.5 ppm for each assignment) and most of the ¹³C chemical shifts overlap. To avoid confusion, we have now changed the sentence on line 277 to “there are no distinct ¹³C chemical shifts corresponding to form C (Figs. 4a and b)^{42,43}”.

4. At line 245-247 “In the ¹³C and ¹⁵N spectra, two separate peaks from the chemically

equivalent site of form B are attributed to the presence of two magnetically inequivalent molecules in the unit cell. “ No clear indication has been provided on the two separate peaks considered in the ^{13}C and ^{15}N spectra.

Our response:

We thank the reviewer for pointing out this unclear sentence. We have now revised this sentence to more clearly describe which peaks are being referred to.

On line 284:

“Considering the chemical formula ($\text{C}_{10}\text{H}_{16}\text{N}_6\text{S}$), the ^{13}C and ^{15}N spectra contained additional peaks that displayed splitting, such as C1/C11, C5/C15, C6/C16, N3/N9, N4/N10, and N6/N12. These peak splittings might be attributed to the existence of two magnetically inequivalent sites for the chemically equivalent atoms of form B.”

5. At line 296 the authors mention the magenta and sky-blue circles in figure 6b, which are not present. Do the authors refer to figure 6c? It should be clarified or corrected.

Our response:

We apologize for the confusion regarding this description. We have now added colored circles to Figs. 6b and c as intended and corrected the corresponding text on line 350.

“The molecules in conformation I are arranged in a head-to-tail manner owing to the intermolecular hydrogen bonds of N1–H1•••N5 and N3–H3•••N2 between the imidazole ring and the guanidine group, as indicated by the sky-blue solid circles, whereas the molecules in conformation II are stabilized by the intermolecular hydrogen bonds of N9–H9•••N7 and the intramolecular hydrogen bonds of N8–H8•••N11, as indicated by the magenta solid circles (Figs. 6b and c). Furthermore, the layers composed of molecules in conformations I and II are alternately arranged in the order II–I–II along the a axis by the hydrogen bonds of N4–H4•••N12 and N10–H10•••N6, as indicated by the orange solid circles (Fig. 6b and c).”

6. In Figure 6c, “I” and “II” labels would help to understand the description of the molecular arrangement. Also, the numbering of the N and H atoms involved in the bonds described in the main text should be added to facilitate the reader.

Our response:

We thank the reviewer for this helpful suggestion to improve our manuscript. We have now added “I” and “II” labels to Figs. 6b and c and shown the numbering of the N and H atoms involved in hydrogen bonding in Fig. 6c, as explained in the main text. We have also added

the numbering of the atoms involved in hydrogen bonding in Fig 3b.

Reviewer #2 (Remarks to the Author):

This is a nice paper reporting on the combined use of solid-state NMR, electron diffraction, and quantum mechanical calculations of chemical shifts, to determine crystal structures of small molecules. This approach has been dubbed "NMR crystallography" and has become popular in recent years for elucidation of crystal structures, as evidenced by over 200 papers published on the subject in the recent decade or so. The main advantage of combining the different techniques (most commonly, X-ray crystallography, solid-state NMR, and QM calculations) is that proton positions and protonation states can be determined, which are typically not visible in most of the X-ray or ED structures. Hence significant effort has been put into the development of this approach by multiple NMR groups across the globe.

The current paper represents yet another example of such an approach, using a well-studied example of L-histidine whose crystal structures are well known and a cimetidine form B with unknown structure. The authors demonstrate that their approach is viable, using L-histidine, where they delineate the only structure consistent with all of the data. They then apply this strategy to cimetidine form B.

Our response:

We are grateful to the reviewer for these positive comments. We agree on the importance of NMR crystallography and believe that this technique possesses numerous applications.

The main novelty of the work presented by the authors is that, instead of X-ray crystallography, they employ electron diffraction. The rest of the study follows the large body of the previously published literature. The technical quality is overall high. However, all of the concepts, the experimental and computational methods are well established and straightforward. The solid-state NMR data, while of high quality, reproduce the multiple reports from many different groups. There is no new information - either technical or conceptual there.

In this reviewer's opinion, this work is valuable for researchers interested in small-molecule crystal structures, including those of pharmaceutical solids. Since there are no major methodological innovations, the work would be much better suited for publication in a more specialized journal, such as Journal of Physical Chemistry, PCCP or PCP. The authors

should also fix multiple grammatical errors and typos in the revised manuscript.

Our response:

As pointed out by the reviewer, NMR crystallography has recently become very popular. However, the currently accepted approach is based on XRD and its broader application is limited by a major obstacle, namely, in the absence of XRD-quality single crystals or pure microcrystalline powders, XRD and PXRD cannot be used for structure determination. Hence, we developed the current approach by combining ED, NMR, and quantum computation to overcome this obstacle and allow the wider application of NMR crystallography. We have highlighted this point in the Introduction of the revised manuscript on lines 82 to 89. We have also corrected the typographical and grammatical errors (see the comments to reviewers 1 and 3).

On line 82:

“As the information provided by SSNMR is complementary to that obtained using XRD, this approach, which is referred to as NMR crystallography, has become popular for the structural determination of single-crystal or pure powder samples. However, there remain several obstacles that limit the widespread application of this approach. When samples form as nanocrystals or contain multiple components such as in the case of pharmaceutical tablets, it is difficult to solve the molecular structure using XRD, which hampers the NMR crystallography approach.”

Reviewer #3 (Remarks to the Author):

The manuscript describes the structure determination of two organic pharmaceutical compounds with the emphasis on the correct determination of the hydrogen bonding network and correct assignment of potentially ambiguous atomic types. The method used is the combination of 3D electron diffraction with ssNMR and DFT calculations. It is convincingly shown that ssNMR combined with simulations provides an efficient tool to supplement information, which is difficult to obtain solely by 3D electron diffraction.

The novelty of the manuscript is not in the use of ssNMR to complete diffraction data. This is being frequently done, especially for pharmaceuticals. The novelty may be seen in the fact that the same method that is common for x-ray diffraction may be successfully used in combination with electron diffraction to obtain accurate structures from micro- and nanocrystals. This fact as such is not really surprising, but it is certainly useful to prove it and

to alert the scientific community about this option.

Our response:

We are grateful to the reviewer for these positive comments. We also believe that it is important to notify the scientific community of our developed method.

While the general outline of the manuscript is fine, the discussion sound and the results convincing, I got a general impression that, in order to strengthen the importance of the proposed method, the potential and ability of structure solution by electron diffraction is underestimated at several places of the manuscript, either on purpose or due to the lack of experience. I will describe the instances of this problem in more detail below, and I suggest this is corrected, otherwise a false impression of the potential of ED will be conveyed.

Our response:

We thank the reviewer for these valuable suggestions. We do not wish to create a false impression of the potential of ED and we have attempted to revise the manuscript following the reviewer's suggestions (see below).

Another problem is that the manuscript would strongly benefit from proof reading focused on the correction of language and wording.

Our response:

We have revised the manuscript thoroughly and also submitted it to a proofreading service to further correct any remaining errors and improve the overall readability of the manuscript (see the comments to reviewer #1).

Specific comments:

General note: the word precision is used often in the manuscript, but possibly not always in the correct meaning. The word precision means "small statistical error". Often the more appropriate word is "accuracy", which means "deviation from the correct value".

line 47: "...with high precision." What is meant by "high precision? In general the hydrogen positions need not be known to very high accuracy to understand the hydrogen bonding network.

Our response:

We apologize for the confusion due to our poor choice of terminology and appreciate the valuable comments. In the revised manuscript, we have carefully corrected the usage of the terms "accuracy" and "precision". We have also conducted a detailed analysis of the accuracy and precision of the hydrogen atom positions using quantum computation and

SSNMR and added a discussion of this to the main text and Supplementary Information. Please refer to our responses below.

lines 88-91: the same problem as in previous comment. The accuracy of the hydrogen positions was about 0.1 Angstroms or better in the cited reference. This is sufficiently high accuracy to determine the hydrogen bonding network and to distinguish salt from cocrystal. It is not important to determine the hydrogen position to a high accuracy, but with high reliability, i.e. be sure that we see all hydrogens in their correct places. The cited work, but also other works (Clabbers et al., *Acta Crystallogr. A* 2019, 75, 82–93; Hynek, J. et al. *Angew. Chem. Int. Ed.* 2018, 57, 5016–5019.) show that this is quite well possible in many cases.

Our response:

As pointed out by the reviewer, there have been several previous reports concerning the use of ED to successfully locate hydrogen atoms with relatively high accuracy. However, this is not always the case. We have attempted to address this problem and provide a general solution in the current manuscript. In a previous paper studying the metal–organic framework ICR-2 (Hynek, J., et al., *Angew. Chem. Int. Ed.* 2018, 57, 5016), only seven of the fifteen hydrogen atoms were observed, despite the use of dynamical refinement. In other words, eight hydrogen atoms were missing. Likewise, if the system is susceptible to strong electron beams or very complex (Dai, R., et al., *Inorg. Chem.* 2017, 56, 8128), it may be difficult to determine all of the hydrogen atom positions even with advanced dynamical refinement. Although this approach is becoming more straightforward, additional calibration and examination are still often required depending on the crystal shape and complexity and an initial model is also needed. Furthermore, the reported studies used a highly sensitive hybrid pixel detector to boost the very weak diffraction from hydrogen atoms. Unfortunately, such detectors are not always available. Therefore, when dynamical analysis or the use of superior equipment is impractical or impossible, SSNMR may represent an alternative method for complete structure determination. In addition, although ED has the capability to elucidate hydrogen-bonding networks, far higher accuracy than that accessible using ED (0.1 Å) is still required in the fields of crystal engineering, medicine, and materials science to evaluate hydrogen-bonding strengths. To convey the intended message to the reader and clarify the importance of the accurate determination of hydrogen atom positions, we have now revised and added the following text on line 97:

“In these ideal cases where all of the hydrogen atoms are visible in the ED potential maps, the hydrogen atoms can be located with sufficient accuracy to understand the hydrogen-bonding network. However, even with this advanced approach, dynamical

refinement does not always permit localization of all of the hydrogen atoms and moreover is not applicable to samples susceptible to radiation damage and/or possessing complex structures^{23,24}.”

line 135: the difference between 22.71 and 23.86 seems relatively large. The claim that electron diffraction is unable to distinguish the two orientations is thus misleading. Moreover, the distinction can be also made by inspection of the displacement parameters and/or difference Fourier map. My suspicion is that these additional checks would further support the estimation made from the *R*-values, making the distinction between the two structures almost unambiguous. It would be more appropriate to say that the distinction may not be 100% certain and may need confirmation. However, the general claim that the distinction of atoms with close atomic numbers may be difficult, is correct, unless dynamical diffraction effects are taken into account, in which case ambiguities are very rare.

Our response:

We agree that the difference of 1.15% in the *R* factors is possible to distinguish LH1(2) and LH3(4). The *R* factors of 22.71 for LH1(2) and 23.86 for LH3(4) were calculated prior to placement of the hydrogen atoms. After the addition of the hydrogen atoms, the *R* factors decreased to 20.09 (LH1), 20.41 (LH2), 21.28 (LH3), and 21.21 (LH4). Therefore, the ED analysis may exclude the possibility of LH3(4) but is not sufficient for distinguishing LH1 and LH2. As recommended by the reviewer, we evaluated the ADP (atomic displacement probability) values and difference Fourier map (Supplementary Figure 1). The ADPs drawn at 50% probability level revealed that the large ADPs on the nitrogen atoms of LH3(4) could exclude the LH3(4) models from being the correct structure. However, both models exhibited significant variation of the ADPs from atom to atom. Thus, we consider that the distinction between these two structures remains unclear. Furthermore, the difference potential maps failed to reveal all of the hydrogen atom positions for both models (LH1(2) and LH3(4)). These results clearly demonstrate the necessity of SSNMR measurements to confirm the correct structure. We have added a discussion of these points to the main text and revised Supplementary Figures 1 and 11.

On line 150:

“Although the larger *R* factor and abnormal atomic displacement probability (Supplementary Figure 1 and Fig. 2b) for the LH3(4) models could exclude the possibility of these being the correct structure, more definitive confirmation was needed. The difference of 1.15% in the *R* factors of LH1(2) and LH3(4) indicates the limitation of ED in distinguishing atoms with similar atomic numbers.”

line 154: the supplementary figure 1 is used to support the claim that hydrogen positions cannot be found with sufficient precision. However, the figure contains the total potential map with relatively high isosurface level, but the hydrogen positions (and other weak features) are best found in a difference potential map (as e.g. in ref. 13). The claim on line 154 should be supported by a difference potential map showing no or only a few maxima corresponding to hydrogen positions.

Our response:

We agree with the reviewer's valuable suggestion and believe that it greatly strengthens the discussion. We have now replaced Supplementary Figures 1 and 11 with difference potential maps ($F_o - F_c$) showing the maxima. Although the difference potential maps revealed several possible hydrogen atom positions, the bond lengths or bond angles were not sufficiently reliable for structure refinement. As stated in the earlier response, we agree that ED possibly allows LH1(2) and LH3(4) to be distinguished, but it remains difficult to distinguish LH1 and LH2 because the differences between these two structures originate solely from the hydrogen atom positions. We have now revised this description in the manuscript.

On line 173:

"Although the difference potential maps were not sufficient for determining all of the hydrogen atom positions (Supplementary Figure 1), this large decrease in the R factors demonstrates the importance of the presence of the hydrogen atoms during the structure refinement process. Nevertheless, the R factors did not reveal whether LH1 or LH2 was the correct structure even if LH3 and LH4 were excluded."

line 167: please specify the standard uncertainty of the bond lengths. The high accuracy of the hydrogen structure determination by ssNMR is a central topic of the paper and the precision of the bond length determination should thus be carefully computed and given in the manuscript.

Our response:

As pointed out by the reviewer, it is of central importance to include the precision and accuracy of the bond lengths reported in this paper. To answer the reviewer's question, we have carefully conducted experiments, calculations, and analysis related to the bond lengths.

First, we performed a set of GIPAW calculations for L-histidine (LH1) to verify the precision of the quantum calculations for various pseudopotentials. We did not observe a large

dependence of the bond length on the choice of pseudopotential and the precision was on the order of 0.01 Å (see Supplementary Figure 5 and the Methods section of the manuscript).

We have now added the following discussion to line 233:

“We next performed a set of GIPAW calculations on LH1 at various pseudopotentials to understand the statistical variation of bond lengths. The N3–H3 bond length was not largely dependent on the pseudopotential and the precision of the quantum computation was on the order of 0.01 Å (Supplementary Figure 5).”

Next, we evaluated the standard uncertainty of the bond lengths measured using SSNMR. This was calculated from the invCP-VC spectra using the full width at half-maximum of the peaks in slices extracted along the dipolar coupling dimension. The standard uncertainty ranged from ± 0.01 Å to ± 0.07 Å. It should be noted that the experimental error calculated from the full width at half-maximum of these peaks was dominated by the spinning rate, and thus could be further improved by increasing the MAS rate. A previous work confirmed that the peak position at maximum remains constant (Paluch, P. et. al., *J. Mag. Reson.* 2013, 233, 56). This range of standard uncertainty is quite comparable to the results obtained using other methods, such as X-ray and ED with dynamical refinement (Clabbers, et al., *Acta Crystallogr. A* 2019, 75, 82; Hynek, J., et al., *Angew. Chem. Int. Ed.* 2018, 57, 5016). The standard uncertainty for each bond length has been added in parentheses on lines 188 and 316 and Supplementary Tables 5 and 9.

To further evaluate the bond lengths, we conducted a set of GIPAW calculations in which certain X–H bond lengths were varied. For each distance, the calculated and experimental NMR chemical shifts were compared. We must admit that this method is not well established, although we believe that the results provide some guidance for evaluating the accuracy of the bond lengths. Note that a similar approach was previously demonstrated in 2003 (Harris, R.K., et al., *ChemComm* 2003, 2834). The obtained results have been included in Supplementary Figures 6 and 15. For instance, in Supplementary Figure 6, the chemical shifts in the first column were obtained by varying only the N3–H3 bond length. Whereas the ^{15}N and ^1H chemical shifts varied, only minor changes were observed in the ^{13}C chemical shifts. This was somewhat predictable as we only varied the N–H bond length. Variation of the C–H bond lengths only led to changes in the ^1H shifts. The bond lengths at the energy minima (GIPAW calculations), at the chemical shift RMSD minima, and derived from SSNMR (red broken line) were in good agreement (± 0.04 Å), indicating good accuracy.

It is interesting to note that the bond lengths derived from SSNMR were consistently 0.02–0.03 Å longer than those at the energy minima. This is a well-known effect of dynamical averaging. While the GIPAW calculations were performed at 0 K, the SSNMR measurements were conducted at room temperature. Thus, the dipolar couplings, from which the SSNMR distances were derived, were dynamically averaged. As the distance is inversely proportional to the cube root of the dipolar interaction, this leads to SSNMR distances that are several percent longer than the average distances, as demonstrated in previous studies (e.g., Ishii, Y., et al., *J. Chem. Phys.* 1997, 107, 2760; Dračinský, M., et al., *CrystEngComm* 2013, 15, 8705).

Similarly, for cimetidine form B (Supplementary Figure 15), the bond lengths at the energy minima, at the chemical shift RMSD minima, and derived from SSNMR were consistent with each other. The bond lengths derived from SSNMR were again approximately 0.01–0.03 Å longer than those at the energy minima owing to the influence of dynamical averaging. Therefore, the consistency of the bond lengths obtained from the energy minima, chemical shift RMSD minima, and SSNMR experiment indicate that these bond lengths were reasonably accurate.

We have now added the following discussion to lines 236 and 329:

“To evaluate the SSNMR-derived bond lengths, we further calculated the chemical shifts and energies as functions of the N3–H3, C2–H2, C5–H5, and C6–H6 bond lengths (Supplementary Figure 6). In all cases, the bond lengths at the energy minima were 0.02–0.03 Å shorter than the SSNMR-derived bond lengths. This is consistent with the well-known relationship that SSNMR bond length based on the size of internuclear dipolar interactions are several percent longer than the distance averaged from atomic positions owing to the influence of dynamical averaging^{35,36}. This suggests that the SSNMR-derived bond lengths are reasonably accurate.”

“We further evaluated the SSNMR-derived bond lengths by calculating the chemical shifts and energies as functions of the N1–H1, N3–H3, N4–H4, N8–H8, N9–H9, and N10–H10 bond lengths (Supplementary Figure 15). All the N–H bond lengths were 0.01–0.03 Å shorter than the SSNMR-derived bond lengths owing to the effect of dynamical averaging as described for l-histidine. The consistency of the bond lengths obtained from the GIPAW calculations and SSNMR experiment indicated that the SSNMR-derived bond lengths were reasonably accurate.”

line 168: it appears to me that the authors are comparing the calculated bond lengths with the default values in SHELXL for x-ray diffraction data. If this is the case then it is not correct, because the default values for x-ray data are shorter than the real distances due to the shift of the hydrogen's electron to the covalent bond. It is not appropriate to use these values for electron diffraction. The comparison should be done with respect to some accepted source of tabulated X-H distances.

Our response:

We had no intention to stress the difference between the results of the riding model in SHELXL and the SSNMR-derived bond lengths. Therefore, to avoid confusion, we have removed the text "which are longer than the default lengths given by SHELXL" from line 189.

line 173: please give the values for R-values for all four structures. The range 19.81 - 21.76 appears relatively large to be classified as "very similar" - see my comment for line 135.

Our response:

We have provided all of the *R* factors for all of the models of both L-histidine (line 194) and cimetidine (line 320) and deleted the word "very" as shown in the sentences below:

On line 194:

"After adjusting the weights during the final step of structure refinement, the *R* factors decreased to 19.81%, 20.00%, 21.06%, and 21.76% for the LH1, LH2, LH3, and LH4 models, respectively."

On line 320:

"weight adjustment afforded similar final *R* factors of 19.96%, 19.69%, 20.13%, and 19.87% for the CB1, CB2, CB3, and CB4 models, respectively, which did not permit determination of the correct structure."

line 269: here again the difference potential map should be shown to support the inability of ED data to locate hydrogen atoms.

Our response:

We have replaced Supplementary Figure 11 with a difference potential map ($F_o - F_c$) showing the maxima. The non-hydrogen atoms are represented using the ADP (50% probability). The difference potential map for cimetidine form B was better than that for L-histidine because the maxima at N1, N9, N10, C3, and C17 revealed possible hydrogen atom positions. Nevertheless, the difference potential map still did not reveal the positions of all of

the hydrogen atoms.

We have revised the sentence on line 308 as shown below:

“In the difference potential map, most of the hydrogen atoms could not be placed automatically (Supplementary Figure 11). Although the difference potential map revealed several possible hydrogen atom positions, unreasonable bond lengths or bond angles prevented further refinement.”

Supplementary table 9: please indicate standard uncertainties of the ssNMR bond lengths in the table.

Our response:

We have added the standard uncertainties of the SSNMR-derived bond lengths to Supplementary Tables 5 and 9. These values are shown in parentheses in the SSNMR column and were calculated using the full width at half-maximum of the peaks in slices extracted along the dipolar coupling dimension. To avoid overcomplicating the discussion in the manuscript, we have provided the average error values for the N–H bond lengths of the guanidine group (N3–H3, N4–H4, N9–H9, and N10–H10).

On line 314:

“The hydrogen positions were then refined using the internuclear distances measured via SSNMR: C–H distance = 1.12 Å (± 0.01 Å) for the imidazole ring and 1.06 Å (± 0.01 Å) for C5–H5, C6–H6, C7–H7, C15–H15, C16–H16, and C17–H17, and N–H distance = 1.06 Å (± 0.04 Å) for the imidazole ring and 1.04 Å (± 0.06 Å) for the guanidine group (Supplementary Figure 12).”

REVIEWERS' COMMENTS:

Reviewer #1 (Remarks to the Author):

All the points raised by the referees have been taken into consideration and a precise discussion has been provided by the authors. The work has been modified accordingly. In conclusion I believe that the paper is innovative, well written, and the provided evidences are convincing. So in my opinion the paper deserves publication on an important journal such as Nature Communication.

Reviewer #3 (Remarks to the Author):

The authors have addressed all my comments in detail and with rigor. The manuscript has been considerably improved. It can be published in the current version.

REVIEWERS' COMMENTS:

Reviewer #1 (Remarks to the Author):

All the points raised by the referees have been taken into consideration and a precise discussion has been provided by the authors. The work has been modified accordingly. In conclusion I believe that the paper is innovative, well written, and the provided evidences are convincing. So in my opinion the paper deserves publication on an important journal such as Nature Communication.

Reviewer #3 (Remarks to the Author):

The authors have addressed all my comments in detail and with rigor. The manuscript has been considerably improved. It can be published in the current version.

Our response:

Thank you very much for positive comments. Again we would like to thank to all the reviewers for the valuable inputs.